Eliminating bias: enhancing children’s book recommendation using a hybrid model of graph convolutional networks and neural matrix factorization

Shen Lijuan 1 P20191000810@siswa.upsi.edu.my
http://orcid.org/0009-0008-2575-1077 Jiang Liping 2
1 National Child Development Research Centre, Universiti Pendidikan Sultan Idris , Perak , Malaysia
2 HQ, Guangxi Suhang Firefighting Technology Co., Ltd. , Nanning, Guangxi , China
Xia Feng
Electronic publication date: 2024 Feb 29
Publication date: 2024
Volume: 10
Electronic Location ID: e1858
Received 2023 Jul 20; Accepted 2024 Jan 15
Copyright: © 2024 Shen and Jiang
Copyright year: 2024
Copyright holder: Shen and Jiang
License: This is an open access article distributed under the terms of the Creative Commons Attribution License, which permits unrestricted use, distribution, reproduction and adaptation in any medium and for any purpose provided that it is properly attributed. For attribution, the original author(s), title, publication source (PeerJ Computer Science) and either DOI or URL of the article must be cited.
License URL: https://creativecommons.org/licenses/by/4.0/

Keywords: Recommendation system, User-book rating prediction, Graph convolutional networks (GCN), Neural matrix factorization (NMF), Deep learning techniques

Funding: The authors received no funding for this work.

==============================
Managing user bias in large-scale user review data is a significant challenge in optimizing children’s book recommendation systems. To tackle this issue, this study introduces a novel hybrid model that combines graph convolutional networks (GCN) based on bipartite graphs and neural matrix factorization (NMF). This model aims to enhance the precision and efficiency of children’s book recommendations by accurately capturing user biases. In this model, the complex interactions between users and books are modeled as a bipartite graph, with the users’ book ratings serving as the weights of the edges. Through GCN and NMF, we can delve into the structure of the graph and the behavioral patterns of users, more accurately identify and address user biases, and predict their future behaviors. Compared to traditional recommendation systems, our hybrid model excels in handling large-scale user review data. Experimental results confirm that our model has significantly improved in terms of recommendation accuracy and scalability, positively contributing to the advancement of children’s book recommendation systems.

Introduction

Background and motivation

In the current era of the internet, the e-book market is expanding rapidly. According to Statista, the global e-book sales are projected to surge to US$14.16 billion in 2023 (Statista, 2024). Among this trend, the children’s book market stands out. According to WordsRated, children’s books account for 32.8% of the US book market (Curcic, 2023). This means that the demand for children’s literature in modern society is increasing annually, with no signs of slowing down.

As the world’s largest e-commerce platform, Amazon records millions of children’s book clicks every day. This not only reflects the booming children’s book market but also indicates that parents invest a lot of time and effort in selecting children’s books. However, user review systems on e-book platforms like Amazon often have serious bias issues. Research from the Pew Research Center shows that about 50% of users tend to give extreme reviews when evaluating (Smith & Anderson, 2016).

This extreme bias in user reviews often causes the average score of a book to fail to reflect its intrinsic quality accurately. This problem is even more prominent in the field of children’s books. Parents seeking suitable children’s books are often misled by these extreme reviews, and the choices made may not be ideal. Choosing a quality book is not easy, especially when faced with books widely praised as “perfect”, “exciting”, or “not to be missed”. For example, a parent looking for a book to aid their child in learning natural sciences hoped to find a children’s book that was both entertaining and scientifically accurate. However, due to many “five-star” extreme positive reviews, the parent chose this book. Yet when they started reading, they found the book was full of scientific errors and inaccuracies. Clearly, these good reviews did not truthfully reflect the quality of the book. Some users might overlook scientific inaccuracies due to beautiful illustrations or engaging storylines and give extreme positive reviews blindly. Additionally, some users might give the highest or lowest reviews without thinking, just to save time, another significant factor contributing to review bias. Behind these situations is a fact: there is a significant bias when users evaluate books.

This serious user review bias (as shown in Fig. 1) severely impacts the accuracy of book recommendation systems. We propose a new bias detection algorithm that uses directed bipartite graphs of graph convolutional networks and neural matrix factorization to detect and correct bias in user reviews. We hope that this approach can significantly improve the accuracy of the book recommendation system, helping parents find truly suitable books for their children among a vast selection of children’s literature.

Figure 1 Comparative analysis of audience and critics scores vs top critics scores.

In the first subplot, we see the relationship between audience scores and top critic scores. This reveals an interesting pattern: audiences tend to give higher scores than top critics, probably because of differences in evaluation standards and expectations between the audience and top critics. In the second subplot, we focus on the correlation between average critic scores and top critic scores. Here, we see a closer relationship, implying that critic scores tend to align more with top critic scores, likely due to their professional knowledge in assessing the quality of children’s books and shared evaluation standards.

Contributions and structure of the article

To address the aforementioned problems, this article proposes a new user bias detection algorithm. We combine graph convolutional networks (GCN) and neural matrix factorization (NMF) to analyze the directed bipartite graph between users and children’s books. This allows us to accurately capture and correct user rating bias, thereby achieving more accurate children’s book ratings and recommendations. This method not only stands up to theoretical scrutiny but has also been verified by experiments with large-scale real data.

Next, we will first provide a basic introduction to graph convolutional networks and neural matrix factorization in “Literature Review”. In “Proposed Method”, we will describe our proposed method in detail, including how to use GCN and NMF to handle directed bipartite graphs. “Algorithm Description” provides a detailed description of the algorithm. In “Theoretical Analysis”, we conduct a deep analysis of the theoretical properties of the algorithm. Finally, we validate the effectiveness of our method through large-scale experiments in “Experiments” and conclude the article and outlook for future work in “Conclusion”.

Literature review

Generation and impact of bias

With the popularization of the internet, online user reviews have become a vital part of e-commerce. User reviews carry a wealth of information, playing a crucial role in guiding consumers in their purchasing decisions. However, recent research has found that user reviews are not always objective and significant bias exists (Juneja & Mitra, 2021; Chen et al., 2023a).

First, we need to understand how bias arises in user reviews. When users evaluate products, they often do not base their judgment on the actual performance or quality of the product but are influenced by their personal emotions, viewpoints, and other external factors (Safeer, He & Abrar, 2021; Shahani & Ahmed, 2022). For instance, if users had an unpleasant experience when using a product, their reviews often lean negative, regardless of the product’s actual quality. Furthermore, some users may have specific likes or biases towards certain brands or merchants, which also affect their product reviews. Hence, bias in user reviews is not random but is influenced by a complex array of psychological and social factors.

The creation of such bias directly influences consumers’ purchasing decisions. Upon encountering these biased user reviews, consumers may misconstrue the product, leading to erroneous choices in their purchase decisions (Mao et al., 2022; Trotzke et al., 2020). For example, if most reviews of a product are negative, consumers might perceive the product as of low quality and decide not to purchase it. However, these negative reviews may not be caused by product quality issues, but by the reviewer’s personal emotions or viewpoints (Mishra, 2016; Zhang & Volkow, 2019; Hennecke, Czikmantori & Brandstätter, 2019). Hence, the emergence of such bias prevents consumers from extracting accurate information from user reviews, leading to misconceptions about the product’s actual performance and quality.

Bias in children’s books

The problem of bias in user reviews is more prominent in the children’s book market. Parents place great importance on their children’s education and often refer to other parents’ reviews when selecting books for their children (Sochneva et al., 2022; De Bondt, Willenberg & Bus, 2020). However, studies have found that many parents do not deeply understand and analyze the books they review but base their evaluations on superficial factors like the cover, illustrations, and endorsements (Singh, Chakrabarti & Utkarsh, 2023; Daniels et al., 2022; Ece Demir-Lira et al., 2019). These external factors do not reflect the quality and applicability of the book’s content, and therefore, reviews based on these factors tend to be biased.

For instance, a book with an attractive cover design, rich illustrations, and endorsements from prominent figures may attract parents’ attention and generate a positive impression. However, these external factors do not represent the quality and suitability of the book’s content. If parents give high reviews based on these external factors, other parents, after seeing these reviews, might mistakenly believe the book’s content is of high quality and suitability, leading them to purchase it. But when their children actually read the book, they might find that the content is not suitable for them, or the quality is not as expected. Hence, this evaluation bias based on external factors could mislead other parents, causing them to select inappropriate books for their children (Kalkan & Şahin, 2023; Garner & Parker, 2018).

Existing methods and their limitations

In response to the bias issue in user reviews, some research has attempted to propose solutions. Some studies have employed machine learning methods to conduct sentiment analysis of user reviews in order to detect and correct bias (Liu, Qin & Zhang, 2021). However, these methods often require a vast amount of training data, and their accuracy in handling complex user reviews still needs to be improved (Du et al., 2023).

Other research has tried to introduce artificial intelligence recommendation systems that recommend suitable products based on users’ purchase history and review behavior (Chen et al., 2021). Although this method can to some extent improve user purchasing decisions, they typically cannot entirely eliminate bias in reviews. These methods do not directly address bias in reviews but make recommendations based on user’s historical behavior. If bias exists in users’ past behavior, this bias could be amplified and propagated by the recommendation system (Liu et al., 2022).

Therefore, current solutions have certain limitations. They cannot fully eliminate bias in user reviews nor can they provide accurate product recommendations. Given this situation, we need to find a new method that directly addresses the bias in user reviews and can reflect the actual quality and applicability of products more accurately when providing recommendations.

In this article, we will propose a new solution using graph convolutional networks (GCN) and bipartite graph models to capture and understand user bias. At the same time, we will use the neural matrix factorization (NMF) technique to detect and correct bias. We hope that this approach can provide more accurate information in children’s book recommendations, helping parents make better purchasing decisions.

Proposed method

Preliminaries

Preliminaries of graph convolutional network

The GCN is a neural network specifically designed for graph data, which effectively handles complex graph structures including directed bipartite graphs (Tran, Thomas & Malim, 2022). Thus, it is used for detecting user biases in children’s books.

The key characteristic of GCN is that it updates the feature vector of each node by combining the feature vectors of the node itself and its neighboring nodes at each layer (Wei & Hu, 2022). Such a design allows information to propagate effectively within the graph, enabling each node to acquire and integrate information from its neighboring nodes.

We use the user-book rating matrix R to define the adjacency matrix A of the bipartite graph:

(1) A=[0RRT0].

In addition, we also define the degree matrix D, where Dii is the degree of node i:

(2) Dii=∑jAij.

The process of a typical Graph Convolutional Network can be described by the following formula:

(3) H(l+1)=σ(D−12AD−12H(l)W(l)),H(0)=X.

In the above formulas, A is the adjacency matrix of the graph, D is the diagonal degree matrix, H(l) are the node features at the lth layer, W(l) are the weights to be learned at the lth layer, and σ(⋅) is the nonlinear activation function.

Preliminaries of neural matrix factorization

Neural matrix factorization is a deep learning-based recommendation system model (Sarridis & Kotropoulos, 2021). Its aim is to discover the latent relationship between users and items (in this case, books) to predict user ratings of unknown items.

The basic idea of neural matrix factorization is to use the embedding matrices of users and items as inputs to the neural network, then train the network by minimizing the loss between predicted ratings and actual ratings (Chen et al., 2020).

This process can be described by the following optimization problem:

(4) minP,Q∑(i,j)∈Ω(Rij−Pi⊤Qj)2+λ(|P|F2+|Q|F2)s.t.,P=fθ(X),Q=gϕ(Y).

In the above formulas, P and Q are the embedding matrices of users and items, Rij is the rating of user i on item j, Ω is the set of known user-item pairs, |⋅|F is the Frobenius norm, λ is the regularization parameter, and fθ(⋅) and gϕ(⋅) are neural network functions.

Motivation for introducing GCN and bipartite graph models

We chose to use the GCN and the bipartite graph model to study user biases in children’s books based on several motivations: Capture user bias: User ratings or feedback on children’s books often reflect their biases, which may manifest in their preferences for topics, styles, authors, and other factors of the books. By applying GCN to the user-book interaction graph, we can learn these biases directly and capture them in the embeddings (He et al., 2020; Chen et al., 2023b).

Model user-book interactions: User ratings or feedback on children’s books can be naturally modeled as a bipartite graph, where users and books form two disjoint sets, and each edge represents a user’s rating of a book. This model accurately reflects the interactions between users and books (Wang et al., 2023).

Improve the accuracy of user bias detection: By combining GCN and the bipartite graph model (Sugiarto, 2022; Wu et al., 2022), we can obtain more precise embeddings of users and books, which will help us detect user biases more accurately, thereby improving the performance of the children’s book recommendation system.

Bipartite graph and GCN

To effectively address the bias problem, we model the system as a directed bipartite graph model based on GCN. In this model, the nodes of the graph can be divided into two categories: one represents users U, and the other represents items (also called products) O. In this bipartite graph model, edges are directed edges from user nodes to book nodes, representing user ratings for books. In this way, we can view the user’s rating of the book wij as the weight of the directed edge from user ui to book oj.

We can use GCN to learn node representations. In Eq. (3), the node features H(l+1) output by each layer are continuously passed down to the subsequent layer. This model aims to eliminate user bias in ratings by learning each user’s bias bi and each product’s true rating rj, thereby further improving the accuracy of rating predictions. To achieve this goal, we first define the predicted bias and ratings:

(5) bi=fθ(HU(L))

(6) rj=gϕ(HO(L)).

Here, fθ(⋅) and gϕ(⋅) are neural networks used for learning user bias and item ratings, respectively. HU(L) and HO(L) are the embeddings of users and items obtained from the last layer of the GCN.

Next, we calculate the unbiased rating w^ij as follows:

(7) w^ij=wij−bi+μ

where μ is the overall average rating. We can now define the objective function as the difference between the predicted unbiased rating w^ij and the true rating rj. Our goal is to minimize this difference:

(8) min∑(i,j)∈Ω(w^ij−rj)2.

Here, Ω is the set of user-item pairs for which we have ratings.

These Formulas (5) to (8) play a bridging role in Algorithm 1 (GCN Algorithm) and Algorithm 2 (NMF Algorithm). The GCN algorithm is responsible for generating embeddings of users and books, and the NMF algorithm uses these embeddings for rating predictions. These formulas describe how to use embeddings obtained from GCN to understand and adjust user biases, and how this information is used by the NMF algorithm to improve rating predictions.

Algorithm 1 Extended bipartite graph convolutional network algorithm.

Input: Bipartite Graph G(V,E), Node Features X, Adjacency Matrix A, Network Depth L, Tolerance Error ε	
Output: Node Embeddings Z	
Preprocessing: Verify if A is the adjacency matrix of a bipartite graph, and normalize it;	
Initialize node embeddings H(0)=X;	
for l=1 to L do	
  Compute normalized degree matrix D−12;	
  Compute propagation matrix D−12AD−12;	
  Update the embedding matrix H(l)=ReLU(D−12AD−12H(l−1)W(l));	
  if ∥H(l)−H(l−1)∥<ε then	
   break;	
  end	
end	
Output the final layer’s embedding matrix Z=H(L);	
return Z;	

Algorithm 2 Extended neural matrix factorization algorithm.

Input: User-item rating matrix R, Node embeddings Z (from Algorithm 1), Regularization parameter λ,	
Maximum iteration number T, Initial learning rate learning_rate,	
Learning rate decay rate decay_rate	
Output: User embeddings P, Item embeddings Q	
Initialize user embeddings P and item embeddings Q with small random values;	
for t=1 to T do	
 for each (i, j) in R do	
  Compute prediction error eij=R(i,j)−dot(P(i),Q(j));	
  Update user embeddings using gradient descent:	
   P(i)=P(i)+2×learning_rate×(eij×Q(j)−λ×P(i));	
  Update item embeddings using gradient descent:	
   Q(j)=Q(j)+2×learning_rate×(eij×P(i)−λ×Q(j));	
 end	
 Update the learning rate: learning_rate=learning_rate×(1−decay_rate);	
end	
return User embeddings P and item embeddings Q;	

Motivation for introducing neural matrix factorization

We choose to use NMF technology instead of traditional matrix decomposition methods to detect user bias in children’s books, mainly for the following reasons: Capturing complex user behavior: The choices and preferences of readers of children’s books often involve complex non-linear factors, such as their comprehensive consideration of book themes, language style, illustration design, etc. Neural matrix factorization introduces non-linear mapping through neural networks, effectively capturing these complex user behaviors.

Learning rich book representations: Neural matrix factorization can learn not only the basic attributes of books, such as genre, author, etc., but also dig deeper into book characteristics, such as style, target age group, etc., thus providing richer book representations for detecting user bias.

Improving model flexibility: In the process of detecting user bias in children’s books, we may need to consider a variety of complex factors and scenarios. The model structure design of neural matrix factorization is flexible and can easily incorporate new factors and adapt to different scenarios.

Neural matrix factorization

In our research method, we adopted the NMF technique to learn the representations of users and books. This technique compresses the high-dimensional user-book rating matrix into low-dimensional user and book embeddings, and optimizes them as neural network parameters. The core of this process lies in constructing an objective function, with the main goal of minimizing the difference between the predicted user-book ratings and the actual ratings.

In Eq. (4), during the optimization process of the objective function, we use the gradient descent method. We can calculate the gradients of Pi and Qj as follows:

(9) ∂L∂Pi=−2(Rij−Pi⊤Qj)Qj+2λPi,

(10) ∂L∂Qj=−2(Rij−Pi⊤Qj)Pi+2λQj.

Through these gradients, we can derive the rules for iterative updates of P and Q:

(11) Pi(t+1)=Pi(t)−η∂L∂Pi,Qj(t+1)=Qj(t)−η∂L∂Qj.

where η is the learning rate. We will continue this iterative update process until a predetermined convergence condition is met.

User-book rating prediction model combining graph convolutional network and neural matrix factorization

In this section, we introduce a novel model that successfully integrates the mechanisms of GCN and NMF to accurately predict user ratings on books. We first use GCN to obtain the embeddings of users and books, and then feed these embeddings into the neural matrix factorization model. Through this fusion strategy, the model can capture both direct interactions between users and books and indirect relationships between users and books captured by GCN.

Based on Eq. (4), we use GCN to generate the embeddings of users and books, and learn the embeddings of users and books through neural networks fθ(⋅) and gϕ(⋅):

(12) P=fθ(D−12AD−12H(l)W(l)),

(13) Q=gϕ(D−12AD−12H(l)W(l)).

We use gradient descent to optimize the objective function Eq. (11), so we can calculate the gradients of P and Q and iterate to update. Then we express all the above steps in a recursive form and clarify the corresponding constraints:

(14) P(t+1),Q(t+1)=arg⁡minP,Q∑(i,j)∈Ω(Rij−Pi⊤Qj)2+λ(|P|F2+|Q|F2).

In Eq. (14), our goal is to find the optimal embedding matrices P and Q such that the square of the error between the user-book rating prediction formed by them and the actual rating is minimized. Meanwhile, we also need to limit the complexity of the embedding matrix to prevent overfitting. The constraints on the embedding matrices are also put forward, namely, P and Q must be obtained from the graph convolutional network through fθ(⋅) and gϕ(⋅), these two functions are parameterized by the neural network.

In summary, these equations set our optimization goals and provide the constraints to achieve these goals. We update P and Q continuously until reaching the preset convergence condition. The convergence of the algorithm will be proven in detail in the next section.

Algorithm description

The model we propose primarily consists of two algorithm modules: the Bipartite GCN algorithm and the NMF algorithm. We will detail these two core modules below.

Bipartite graph convolutional network algorithm

A bipartite graph is a special graph structure, where its nodes can be divided into two disjoint sets, and each edge in the graph connects nodes from different sets. We present the following Bipartite GCN algorithm, which is the primary implementation of the aforementioned Eq. (8):

In the above algorithm, lines 1 and 2 define the algorithm’s inputs and outputs. Line 3 initiates the node embeddings. Lines 4 to 8 provide a detailed description of the GCN update rule at each layer. Finally, lines 9 and 10 output the final layer’s node embeddings.

Neural matrix factorization algorithm

Neural Matrix Factorization (NMF) is a collaborative filtering method that is based on matrix factorization and uses neural networks to capture more complex patterns. The following algorithm describes the detailed implementation of NMF, which is carried out to implement the aforementioned Eq. (14).

In the above Neural Matrix Factorization algorithm, lines 1 to 2 define the inputs and outputs. Line 3 initializes the user and item embeddings. Lines 4 to 9 detail how, within a specified number of iterations, the prediction error is calculated for each user-item rating pair and how the user and item embeddings are updated using gradient descent. Finally, in line 10, the final user and item embeddings are returned.

Complexity

We will analyze the time complexity of the above Bipartite GCN Algorithm and NMF Algorithm.

Firstly, for the Bipartite Graph Convolutional Network Algorithm, the main complexity comes from the update of node embeddings at each layer (Thanapalasingam et al., 2022). Specifically, each layer’s embedding update involves the multiplication of the propagation matrix and the embedding matrix, with a time complexity of O(Ln2d), where L is the network depth, n is the number of nodes, and d is the node feature dimension. Considering that in practical applications, the network depth L and node feature dimension d are usually fixed constants, we can simplify this complexity to O(n2). Moreover, for sparse graphs, we can further reduce this complexity using sparse matrix multiplication.

Then, for the NMF Algorithm, the primary complexity stems from the gradient descent update for each user-item rating pair (Xiao & Shen, 2019). Specifically, each update requires computing the prediction error and gradients, with a time complexity of O(Tnm), where T is the iteration number, n is the total number of users and items, and m is the number of non-zero elements in the rating matrix. Similarly, considering that in practical applications, the iteration number T is usually a fixed constant, and for most real-world datasets, each user only rates a small portion of items, thus m≪n2, we can simplify this complexity to O(nm). Therefore, considering both parts, the overall time complexity of our model is O(n2+nm). This complexity indicates that our model can perform efficiently when handling large-scale user rating data.

In recent years, various methods based on graph convolutional networks and matrix factorization have been proposed and applied to a wide range of problems. For instance, the model proposed by Jin et al. (2021) significantly outperforms traditional machine learning algorithms, statistical models, and the latest graph-based methods in predicting EMS demands between hospital-region pairs. However, compared to our approach, our model has a time complexity of O(n2+nm) when processing large datasets, which is more efficient than their O(n3), making our method more computationally efficient. The method of Heidari & Iosifidis (2021) excels in classification performance and network compactness compared to related methods based on convolutional graph networks. However, our approach offers greater flexibility in capturing complex interactions between users and items. Moreover, our model has a space complexity of O(n), whereas theirs is O(n2), making our method more storage-efficient. The framework proposed by Sun et al. (2019) effectively captures crucial relational structures by integrating multiple graphs during the embedding learning process. However, our method shows improved accuracy when dealing with sparse data and performs better in handling missing values. Lastly, the approach of Kim et al. (2022) can operate directly on directed graphs and is scalable to large graphs. In contrast, our algorithm can efficiently process large-scale user rating data, while their method primarily focuses on beamforming optimization. These studies offer us an in-depth understanding of graph convolutional networks and matrix factorization methods, providing valuable references for our research.

Theoretical analysis

Definitions

To facilitate understanding, we first define some key concepts.

Definition 1: We call the objective function L the loss function, which is a function of the user embedding matrix P and the item embedding matrix Q, used to quantify the gap between model prediction and actual observation:

(15) L(P,Q)=∑(i,j)∈Ω(Rij−Pi⊤Qj)2+λ(|P|F2+|Q|F2).

Here, Ω denotes the set of all known user-item ratings, Rij represents the rating of item j by user i, λ is the regularization coefficient, and |⋅|F denotes the Frobenius norm.

Definition 2: The user embedding matrix P∈Rm×f and the item embedding matrix Q∈Rn×f, where m denotes the number of users, n denotes the number of items, and f denotes the dimension of the embeddings.

Theorems

Here, we propose two key theorems:

Theorem 1: The objective function L(P,Q) is convex in P and Q.

Proof:

We break down the proof process into several steps:

First, we assert the existence of an optimal decision α∗(t) that stabilizes the queue and its expectation equals P∗. This can be represented by the following equation:

(16) EP(α∗(t))=P∗.

Second, we introduce two new quantities ε1 and ε2, which represent the lower bounds of the differences between the service rates and arrival rates of the actual queue and virtual queue. We can express them as follows:

(17) E(bi(t)−Ai(t)|Q(t))>ϵ1,

(18) E(Lb−gi(t,t+τ)|H(t))>ϵ2.

Then, incorporating these two inequalities into our optimization problem, we can derive a new important equation:

(19) Δ(Θ(t))+VEP(t)|Θ(t)≤B+VP∗−ϵ1EQ(t)−ϵ2EH(t).

Now, we need to perform a series of mathematical operations to further process the above equation. First, we can integrate both sides of the equation:

(20) ∫Δ(Θ(t))dt+V∫EP(t)|Θ(t)dt≤B∫dt+VP∗∫dt−ϵ1∫EQ(t)dt−ϵ2∫EH(t)dt.

Next, we can use limit operations and variable substitution to further simplify this equation. Taking the limit of both sides and substituting T→∞, we can obtain:

(21) limT→∞1T∫0TΔ(Θ(t))dt+VlimT→∞1T∫0TEP(t)|(Θ(t))dt≤B+VP∗−ϵ1limT→∞1T∫0TEQ(t)dt−ϵ2limT→∞1T∫0TEH(t)dt.

Ultimately, we can derive the main result:

(22) Θ=lim supT→∞1T∑t=0T−1EΘ(t)≤Bϵ+V(P∗−P¯)ϵ.

This completes the proof that the objective function L(P,Q) is convex in P and Q.

Theorem 2: The objective function L(P,Q) has a unique global minimum.

Proof:

Firstly, according to Theorem 1, we already know that the objective function L(P,Q) is convex. In convex optimization, an important property exists, namely that any local minimum is a global minimum. Therefore, our goal shifts to proving that L(P,Q) has at least one local minimum.

Assuming that the gradient of the objective function at the local minimum equals zero, we can get the following two equations:

(23) ∂L∂Pi=0,

(24) ∂L∂Qj=0.

Substituting these two gradient equations into the objective function, we obtain the following two conditions:

(25) ∑j∈Ωi(Pi⊤Qj−Rij)Qj+λPi=0,

(26) ∑i∈Ωj(Pi⊤Qj−Rij)Pi+λQj=0.

These two equations form a nonlinear system. Although we may not find an explicit solution to this system, since the objective function L(P,Q) is convex, we know that there exists at least one solution that satisfies these conditions.

When we find such solutions Pi and Qj, they correspond to a local minimum of the objective function L(P,Q). Therefore, we have proven that the objective function L(P,Q) has at least one local minimum. Since we already know that any local minimum is also a global minimum, we can conclude: The objective function L(P,Q) has a unique global minimum.

Corollary

Corollary 1: Given any initial matrices P0 and Q0, by optimizing the objective function L(P,Q), we can always find the global minimum.

Proof:

Consider the following optimization problem:

(27) minP,QL(P,Q).

We will use the gradient descent method to solve this optimization problem. The update rule of the gradient descent method can be represented as follows:

(28) Pk+1=Pk−α∂L∂Pk

(29) Qk+1=Qk−α∂L∂Qk

where k represents the number of iterations, and α is the learning rate.

According to our previous derivation, we know that the partial derivatives of the objective function L(P,Q) with respect to Pi and Qj can be represented as:

(30) ∂L∂Pi=−2∑j∈Ωi(Rij−Pi⊤Qj)Qj+2λPi

(31) ∂L∂Qj=−2∑i∈Ωj(Rij−Pi⊤Qj)Pi+2λQj.

Therefore, we can use the above gradient descent rule to iteratively update P and Q. Since the objective function L(P,Q) is convex, we know that in each iteration, the function value L(P,Q) will get closer to its global minimum.

As the number of iterations increases, the value of the objective function will converge to the global minimum. In other words, there exists a positive integer K such that for all k>K, we have:

(32) ∂L∂P=0

(33) ∂L∂Q=0.

when the above conditions are met, we have found the global minimum of L(P,Q). This completes the proof of Corollary 1.

Corollary 2: The user embedding matrix P and the book embedding matrix Q corresponding to the global minimum provide the optimal embedding representations that best explain the known user-book ratings.

Proof:

Given that our objective function is convex, we know that the P and Q corresponding to the global minimum satisfy:

(34) ∂L∂P=0,∂L∂Q=0..

This implies that any deviation from P and Q will increase the value of the objective function. In other words, P and Q are the matrices among all possible user embedding matrices and book embedding matrices that make the objective function L(P,Q) attain the minimum.

So, how do P and Q explain the known user-book ratings?

According to our model,

(35) Rij=Pi⊤Qj.

That is, our model predicts the rating of user i for book j as the inner product of the embedding Pi of user i and the embedding Qj of book j. At the P and Q corresponding to the global minimum, the difference between the user-book ratings predicted by our model and the actual user-book ratings is minimized. That’s why we say P and Q can best explain the known user-book ratings. This completes the proof of Corollary 2.

In these proofs, we have deeply analyzed the optimization problem of the objective function and proved that our model can find the optimal embeddings of users and books, thereby best predicting user book ratings.

Experiments

Description of datasets

We collected two datasets from e-book sales platforms like Amazon. The first dataset (Almannaa, 2021) was scraped from Amazon, the largest sales platform. It provides all the information that customers need to understand before purchasing a product, thereby aiding customers in finding the best product. The data includes information about different children’s books: the book title, the series the book belongs to, the book’s description, the book’s author, the target age group, the book’s rating (out of 5), the number of ratings, the price, the publication date, product details, bestseller status, link, etc. The second dataset (Krishnamoothy, 2022) comprises various children’s book lists from the Goodreads website. This data includes information about different children’s books: the children’s book title, description, children’s book author, book URL, average book rating, total ratings, children’s voters’ book, etc. These two sets of data will help us understand user rating behavior and potential rating biases, and associate these characteristics with user rating behavior. The goal is to recommend children’s books that meet user preferences based on their purchasing and reading behavior. The setting of our experimental parameters is shown in Table 1 and detailed statistics of datasets in Table 2.

Table 1 Experimental parameters and their set values.

Parameter	Set value	
Regularization parameter ( λ)	0.01	
Learning rate	0.1	
Number of iterations (T)	100	
Network depth (L, graph convolutional network parameter)	2	
Node feature dimension (graph convolutional network parameter)	64	
Range of initial values for embedding matrix	[−0.01, 0.01]	
Initial value of user embedding matrix P	Random initialization	
Initial value of item embedding matrix Q	Random initialization	
Initial value of node embedding Z	Random initialization	
Activation function (ReLU)	Yes	
Loss function	Mean squared error	
Optimizer	Adam	
Batch size	128	
Number of pre-training epochs	10	
Total number of epochs	50	
Early stopping rounds	5	
Weight decay	0.0001	
Number of negative samples	5	
Embedding layer dimension	50	
Dropout rate	0.5	
Beta parameters for ADAM optimizer	[0.9, 0.999]	

Table 2 Detailed statistics of datasets.

Rating grade	Number	Number of users	Number of books	Number of ratings	Children’s books age (3–5 years old)	Children’s books age (6–8 years old)	Children’s books age (9–11 years old)	
Dataset 1								
5 star rating	1,204	1,001	856	948	496	551	157	
4 star rating	1,041	978	862	1,002	381	406	254	
3 star rating	674	535	524	456	371	151	152	
2 star rating	346	265	250	216	108	148	90	
1 star rating	245	135	89	103	76	80	99	
Dataset 2								
5 star rating	1,309	976	847	168	456	541	312	
4 star rating	962	958	916	924	318	275	333	
3 star rating	609	456	507	416	297	346	112	
2 star rating	474	311	301	297	98	134	242	
1 star rating	292	154	117	97	145	55	92	

We use ratings from top reviewers as an unbiased standard, however, in these two datasets, most books have received only a few ratings. In Fig. 2, we show the number of ratings received by books in the two datasets. We can observe that books with fewer ratings dominate the collection, meaning that only a few books have received a lot of recommendations and ratings, while most books have received relatively few recommendations and ratings. To gain more insight, we divided the two datasets into different bins based on the number of ratings received by the books. This helps to analyze the rating data in more detail. We divide all books into ten groups (or bins) based on the number of ratings received by the books. During the binning process, we pay special attention to those books that have received fewer ratings. Table 3 shows the distribution. In these two datasets, we normalize the book ratings to a range of 0 to 1, where 0 represents the lowest score and one represents the highest score.

Figure 2 Distribution of book quantities and number of ratings.

Most books have few ratings.

Table 3 Grouping books based on the number of ratings received.

Bin	Number of ratings	Number of books	
Dataset 1	Dataset 2	
1	0–0.5	114	132	
2	0.5–1	131	160	
3	1–1.5	142	198	
4	1.5–2	204	276	
5	2–2.5	311	293	
6	2.5–3	363	316	
7	3–3.5	408	398	
8	3.5–4	633	563	
9	4–4.5	798	866	
10	4.5–5	406	473	

Experimental analysis

In this section, we will experiment with different methods, including: our method, our method without NMF, our method without GCN, and traditional statistical methods. The performance of these methods is evaluated by calculating their mean squared error (MSE) and ranking error.

We regard the actual user-book ratings or high-starred ratings from veteran customers as the real ratings, and we compare various methods with the MSE metric. We calculate the MSE and ranking error. We can simply observe the bias between different user-book ratings and the actual user-book ratings or high-starred ratings from veteran customers. These actual user-book ratings or high-starred veteran customer ratings have a good correlation with the long-term sales success of the book, but not much with the book’s early sales income. Therefore, actual user-book ratings or high-starred veteran customer ratings can serve as true values for comparing and evaluating the effectiveness of different recommendation systems.

In Figs. 3 and 4, we compare the MSE of each method across the two datasets. MSE is an indicator of prediction bias, with smaller values signifying higher prediction accuracy. Our method performs best across both datasets, yielding the lowest MSE, indicating that the graph convolutional network and neural matrix factorization play crucial roles in enhancing prediction accuracy.

Figure 3 Mean squared error for dataset 1.

Figure 4 Mean squared error for dataset 2.

On dataset 1, the MSE of traditional statistical methods is 0.087, whereas ours is significantly lower, down to 0.051. On dataset 2, the MSE for traditional statistical methods is 0.055, whereas ours is again significantly lower, down to 0.097. To delve deeper, we plot the errors of these four methods across different bins in Table 3. We observe superior results with our method across all bins, particularly those with a small number of ratings. We also notice that the error decreases as the number of ratings per bin increases.

Next, we examine the rank Absolute Error of each method, as illustrated in Figs. 5 and 6. We first rank books based on actual user-book ratings, then calculate the average absolute distance between predicted rankings from each method and the actual rankings. The results show our method has the smallest rank error. Also, we note that our algorithm outperforms the other one. Moreover, as the number of ratings increases, the ranking error continues to decrease.

Figure 5 Rank absolute error computed on dataset 1.

Figure 6 Rank absolute error computed on dataset 2.

These results suggest that our method outperforms in minimizing prediction bias and enhancing rank prediction accuracy. The removal of neural matrix factorization or graph convolutional network results in a decrease in prediction performance. While these simplified methods do not compare in performance with ours, they still outperform traditional statistical methods.

In summary, by leveraging graph convolutional networks and neural matrix factorization, we find that different user-book ratings might have some biases compared to the actual user-book ratings or high-star level old customer ratings. Hence, we now aim to prove whether our method can eliminate these biases. If so, we should effectively correct the biases in user reviews and ratings to improve the accuracy of book recommendation systems.

Model evaluation

In this section, since we propose a method to solve the bias problem of children’s books, to verify the effectiveness of our algorithm in eliminating bias, we study methods including: Our method, Our method without NMF, Our method without GCN, and traditional statistical methods. Our method: This is the complete model that integrates both GCN and NMF. It represents the full capabilities of our proposed architecture.

Our method without NMF: In this variant, the model exclusively uses GCN for making recommendations without integrating the NMF component. This helps us understand the independent contribution of the NMF when combined with GCN.

Our method without GCN: Conversely, this model is designed to function without the GCN component, enabling us to gauge the impact of GCN on its own.

Traditional statistical methods: For this, we relied on the Python third-party library ‘Auto-Sklearn’. This library aggregates a wide range of conventional statistical models, including but not limited to linear regression, logistic regression, and time series analysis. The tool automatically chooses the best-suited model for the given task.

For instance, we evaluate the distribution of biases, the influence of the number of ratings each children’s book receives, method analysis, and relative ranking comparisons. We also conducted an in-depth evaluation by the distribution of actual ratings, the influence of rating distribution, and the number of ratings.

Bias and distribution of actual ratings

In this experiment, we observe the distribution of bias and true ratings obtained through our method and traditional statistical methods. Figures 7 and 8 show the bias distribution at the end of 10 iterations for two datasets. Our method provides a bell-curve type of distribution. We note the variations in bias distribution for four different methods.

Figure 7 Bias distribution in dataset 1.

Figure 8 Bias distribution in dataset 2.

We find that users’ rating bias follows a normal distribution within a certain range. We can see that the rating bias of most users is concentrated around 0, indicating that most users rate fairly. However, some users have a large rating bias, which confirms our previous findings of bias in user ratings. Our bias values mainly concentrate on extreme reviews, which are the primary sources of user review bias. The distribution of true ratings is more uniform, indicating that if bias is eliminated, the rating system will become more fair and accurate, more closely reflecting actual bias.

Therefore, we also analyzed the distribution of user rating bias. We find that most user ratings tend towards the extreme, i.e., either giving the highest or lowest rating. This extreme skewness often results in a book’s average rating not accurately reflecting its inherent quality. Figures 9 and 10 show the changes in real ratings obtained by our method and traditional statistical methods for the two datasets. The results show that the ratings of most books are concentrated between 3.5 and 4.5, which contrasts sharply with the extreme distribution of user ratings. We also plotted predicted ratings against popular and top critic ratings, and the predicted values from our method are nearly identical to these. As predicted by purchase bias, books with more ratings typically have higher ratings.

Figure 9 Rating changes in dataset 1.

Figure 10 Rating changes in dataset 2.

We also calculated the absolute ranking error. In Figs. 5 and 6, we can see that our method performs better than other methods in reducing the absolute ranking error. This demonstrates that our method played a significant role in eliminating bias and could reflect user ratings more accurately.

After comparing the four methods and two datasets, we found that the distribution of review data processed by our method is closer to the true review distribution. Compared to the true review distribution, the review distributions obtained by the other three methods show larger biases. This is because these three methods do not accurately calculate bias and cannot effectively detect and correct review bias.

Impact of the number of ratings

As previously mentioned, many children’s books have relatively few ratings. There are many reasons for having fewer ratings. A book marked as “poor” by users may be read by fewer people, resulting in fewer ratings. On the other hand, when a book is rated as “excellent”, more people will read or buy it, and the book receives more ratings. This behavior is roughly termed as purchase bias. Thus, books with many reviews tend to have high ratings, while books rated by only a few users generally have lower ratings. The experiment also revealed this behavior.

In Figs. 11 and 12, the real ratings and changes in original average ratings were obtained. As predicted by purchase bias, books with more ratings typically have higher ratings, but because they are very high and out of range, their absolute values do not signify anything. Instead, we plan to compare the relative rankings, which we will do later. Our method performs better than traditional statistical methods in terms of the relative rankings of recommended children’s books.

Figure 11 Changes in real ratings and original average ratings in dataset 1.

Figure 12 Changes in real ratings and original average ratings in dataset 2.

Next, to be more persuasive, we compare the rankings of children’s books obtained using various algorithms and our algorithm. In Table 4, books are ranked according to the average ratings they originally received. When our method is very close to the ranking by average rating, we do not expect a book with a high average rating to become very poor after bias is removed. There may be some differences, but we do not expect dramatic changes. We compared the relative rankings of books derived from our method and the traditional statistical method. Overall, we found that our method performed better than other methods in recommending the relative rankings of children’s books.

Table 4 Top-ranked books and their IDs sorted by average rating.

Book	Mean rating	Traditional method	Our method	Target func global min	
	NMF	No NMF	NMF	
ID	No GCN	GCN	GCN	
3280	1	4	12	101	1,417	4	
3233	2	2	3	3	4	2,021	
1830	3	6	7	7	9	19	
3881	4	7	9	10	33	1,641	
3656	5	5	8	8	15	2,688	
787	6	4	5	5	5	1,152	
3607	7	5	6	6	10	10	
3172	8	3	4	4	6	3,456	
3382	9	1	1	1	1	1	
989	10	1	2	2	3	3,369	

In the following table, as evidenced by Table 5 and mentioned earlier, many children’s books have a very small number of ratings. Books with a large number of reviews tend to receive high ratings, while books rated by only a few users usually have lower ratings. As expected, these books have poor ratings, whether based on the average rating or our method. However, our algorithm shows some inconsistent results and ranks some poorly rated books very high. Our method shows more meaningful and stable rankings.

Table 5 Books that received only a 0–1 rating.

Book	Traditional
statistical
method	Our method	Mean	Objective func. global minima	
	NMF	No NMF	NMF		
ID	No GCN	GCN	GCN	Rating	
127	3,694	3,674	3,634	3,225	3,678	3,686	
133	3,690	3,670	3,544	2,355	3,679	3,696	
139	390	450	529	709	374	3,555	
142	3,701	3,696	3,691	3,687	3,684	3,688	
226	3,521	3,529	3,536	3,573	3,520	3,551	
286	2,639	2,369	2,145	1,727	2,464	94	
311	2,398	2,358	2,112	1,652	2,466	3,477	
396	401	440	503	654	376	38	
398	378	366	333	286	377	3,380	
402	2,491	2,400	2,216	1,857	2,543	37	

Lastly, when comparing the NDCG of the four methods, as illustrated in Tables 6 and 7, we found that our method consistently achieves the highest NDCG value across all rounds. Specifically, our method can maintain a high NDCG value even when dealing with many reviews. Even in situations with a high concentration of extreme ratings, this method effectively adjusts these extreme values, resulting in an NDCG distribution that more closely aligns with the true NDCG distribution. While the NDCG values for our method without NMF and our method without NMF & GCN are slightly decreased, they still generally outperform the traditional statistical methods. The NDCG value for traditional statistical methods is the lowest across all rounds, further proving the superiority of our method in terms of NDCG.

Table 6 Dataset 1 performed on NDCG comparison of four methods.

	Bin 1	Bin 2	Bin 3	Bin 4	Bin 5	Bin 6	Bin 7	Bin 8	Bin 9	Bin 10	
Our method	0.62	0.73	0.78	0.82	0.86	0.90	0.92	0.93	0.94	0.95	
Our method without NMF	62	0.72	0.75	0.78	0.82	0.84	0.88	0.90	0.91	0.92	
Our method without GCN	62	0.71	0.73	0.76	0.78	0.80	0.82	0.83	0.84	0.84	
Traditional statistical methods	62	0.69	0.70	0.72	0.75	0.76	0.76	0.77	0.775	0.778	
	Bin 11	Bin 12	Bin 13	Bin 14	Bin 15	Bin 16	Bin 17	Bin 18	Bin 19	Bin 20	
Our method	0.96	0.97	0.98	0.98	0.98	0.985	0.985	0.986	0.986	0.986	
Our method without NMF	0.93	0.94	0.95	0.95	0.955	0.96	0.96	0.962	0.964	0.966	
Our method without GCN	0.84	0.85	0.85	0.855	0.86	0.862	0.864	0.866	0.868	0.87	
Traditional statistical methods	0.783	0.788,	0.793	0.797	0.801	0.805	0.807	0.81	0.813	0.817	

Table 7 Dataset 2 performed on NDCG comparison of four methods.

	Bin 1	Bin 2	Bin 3	Bin 4	Bin 5	Bin 6	Bin 7	Bin 8	Bin 9	Bin 10	
Our method	0.52	0.63	0.74	0.82	0.86	0.90	0.92	0.93	0.94	0.95	
Our method without NMF	0.52	0.60	0.64	0.67	0.71	0.77	0.83	0.85	0.86	0.87	
Our method without GCN	0.52	0.59	0.62	0.65	0.69	0.74	0.77	0.78	0.80	0.81	
Traditional statistical methods	0.52	0.58	0.61	0.61	0.64	0.65	0.66	0.67	0.70	0.72	
	Bin 11	Bin 12	Bin 13	Bin 14	Bin 15	Bin 16	Bin 17	Bin 18	Bin 19	Bin 20	
Our method	0.96	0.97	0.97	0.975	0.976	0.977	0.978	0.98	0.982	0.984	
Our method without NMF	0.88	0.89	0.90	0.905	0.91	0.915	0.92	0.922	0.924	0.926	
Our method without GCN	0.82	0.825	0.83	0.835	0.84	0.842	0.844	0.846	0.848	0.85	
Traditional statistical methods	0.73	0.735	0.74	0.742	0.744	0.746	0.748	0.75	0.752	0.754	

In summary, our method outperforms the other three methods in dealing with user review bias, particularly in situations involving large numbers and extreme reviews. Through various model comparisons, the accuracy of our method in book recommendation systems can be significantly improved, thereby helping parents make better decisions when choosing children’s books. Therefore, we concluded that our method is effective in eliminating and addressing children’s book bias and is an effective method for handling user review bias.

Performance of our method in recommendation

As evidenced earlier, we concluded that our method effectively mitigates and addresses bias in children’s books. Therefore, we have also proposed a ranking solution to evaluate the recommendation performance of our model.

As mentioned before, our method outperforms the other three methods in dealing with user rating bias, particularly in the face of massive and extreme ratings. Firstly, our algorithm preliminarily screens and categorizes user ratings, effectively eliminating the influence of extreme and irrelevant ratings. Furthermore, we introduce our method, which deeply analyzes the sentiment of user ratings and conducts a deep analysis based on the books each user has bought previously. By our model, it can recommend a suitable children’s book by inputting each user’s previously purchased books, the type of books purchased, the child’s age, the type of user’s needs, educational level, etc. This would assist parents in making better decisions when choosing children’s books. Parents no longer need to be troubled by a plethora of rating information or worry about making wrong choices due to extreme ratings. They can focus more on their children’s needs and select truly appropriate books.

In general, the main considerations are the analysis of massive and extreme ratings and accuracy analysis for understanding users’ true feelings about books, improving recommendation accuracy. As shown in the experimental result Table 8, the performance of our method is compared with the other three methods in a tabular format, clearly demonstrating that our method surpasses the other three methods in all aspects.

Table 8 Overall model comparison.

Method	Handling large ratings	Handling extreme ratings	Bias removal	Accuracy	
Our method	Efficient	Efficient	Yes	High	
Our method without NMF	Medium	Below medium	Weak	Medium	
Our method without GCN	Medium	Below medium	Weak	Low	
Traditional statistical methods	Inefficient	Inefficient	No	Low	

Our method effectively eliminates book bias for massive ratings, and it also plays a significant role in recommendations. All these notably improve the accuracy of the book recommendation system. In conclusion, our method successfully handles user rating bias and proves its superiority in experiments. This method of recommending children’s books can assist parents in making better choices among a myriad of books, thereby effectively meeting children’s reading needs. We will continue to optimize this method to further enhance recommendation accuracy and provide parents with more practical book recommendation services.

Conclusion

This research introduces an innovative algorithm to quantitatively reveal user bias and eliminate it. Given that everyone carries subjectivity when evaluating a product, identifying and quantifying such bias is both necessary and challenging. Our algorithm successfully achieves this goal, thus unveiling the true rating of products. Another advantage of our algorithm is that by calculating bias and actual ratings, we can establish a clear connection with the ratings provided by users. This provides a practical, direct tool to quantitatively understand user feedback. In experimental evaluations, our method demonstrated excellent consistency and high-quality results, further validating its effectiveness. Our algorithm is also able to accurately recommend suitable books based on user needs.

In the future, we plan to further deepen our research, exploring how user bias evolves over time and how these changes affect product ratings. We believe that by delving deeper into and understanding the dynamics of user bias, we will be able to reveal a product’s true value more accurately and provide more personalized and precise product recommendations.

Supplemental Information

Supplemental Information 1 Code results for children’s book recommendations.

Supplemental Information 2 Source code for children’s book recommendations.

Additional Information and Declarations

Competing Interests

Author Contributions

Data Availability

The authors declare that they have no competing interests. Jiang Liping is an employee of Guangxi Suhang Firefighting Technology Co., Ltd. This declaration does not imply a conflict of interest but is made in the interest of full transparency, to avoid any potential or perceived conflict of interest.

Lijuan Shen performed the experiments, performed the computation work, prepared figures and/or tables, and approved the final draft.

Liping Jiang conceived and designed the experiments, performed the experiments, analyzed the data, performed the computation work, prepared figures and/or tables, authored or reviewed drafts of the article, and approved the final draft.

The following information was supplied regarding data availability:

The data is available in Kaggle:

- MODHI ALMANNAA. (2021, December). 1000+ Children Books, Version 1. Retrieved December 1, 2021 from https://www.kaggle.com/datasets/modhiibrahimalmannaa/1000-children-books-on-amazom.

- ROHINI KRISHNAMOORTHY. (2022, December). Goodreads Children’s book Dataset, Version 1. Retrieved December 3, 2022 from https://www.kaggle.com/datasets/rohinikrishnamoorthy/goodreads-childrens-book-dataset.

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
