# Peer review of "Eliminating bias: enhancing children’s book recommendation using a hybrid model of graph convolutional networks and neural matrix factorization"

_PeerJ Computer Science, doi:10.7717/peerj-cs.1858_

## Round 0.1 · original submission · Major Revisions

I would appreciate it if the authors could address the issues raised by every reviewer.

A double-check of the whole paper would help as well.

Reviewer 1 ·

Basic reporting

- The usage of metric x in figure 3,4,5,6 can be confusing. Better to use bin x
- Some texts in figure 9 and 10 are cluttered
- Some texts in the table 3 are cluttered
- Better to put basic statistics of each dataset (e.g. how many users, how many book, how many user-book ratings, etc), so that the readers have some intuition regarding the datasets.
- The arrangement of section 2 is confusing:
* On what basis this section is arranged? I think better to arrange this section based on the list of all methods used in the experiments
* The authors use the same title in section 2.1.2 and 2.5. Is it referring to the same thing? why not combining both in the same section then?
* The role of user’s bias and product’s true rating variables in section 2.3 is not clear. Is it incorporated in all models in this paper? or is it only used for the first model? If it is only used in the first model, does it mean the user's bias and product’s true rating were not incorporated in other models?

Experimental design

- The traditional statistical methods are not described clearly
- Both datasets only contain the aggregate ratings of the users, and there is no information about the users. How do the authors construct the adjacency matrix from this information only?
- The assumption behind the choice of the graph embedding dimension is not described
- The context of the use of accuracy metrics in section 5.3 is not described. If it is for ratings prediction, accuracy metric may not be suitable for numerical variables such as ratings. If it is for book recommendation, accuracy is a bad metric because it cannot capture data imbalance and prediction importance. In this case, NDCG should be a better alternative.

Validity of the findings

- The rating prediction of a matrix factorization is basically a dot product between user embedding and book embedding. It means this method is basically learning a GCN embedding through minimization of the difference between rating and dot product, in which most of GCN recommender system already use it as their loss function (e.g. doi.org/10.1145/3397271.3401063, doi.org/abs/10.1145/3543507.3583229, doi.org/10.1016/j.jocs.2022.101855, etc). Therefore, this new method (a combination between NMF and GCN) is actually similar with other common GCNs.

Reviewer 2 ·

Basic reporting

The structure is good, and the language is professional.

Experimental design

The experimental design is well, but has not compare with other works.

Validity of the findings

Impact and novelty not assessed.

Reviewer 3 ·

Basic reporting

The draft is straightforward, self-contained and well-structured. More literature on eliminating bias in recommendations is needed. This is an active research area. However, the literature listed in the draft are not very well-known ones. Besides, the literature reviews, the formulation and writing still need improvement. You may refer to [1]for some literature in this area.
1. In Eq.1, a semicolon is needed or break the line to separate the H^{l+1} and H^{0}
2. If I understand correctly, Eq.6-Eq.9 is not used in your methods, why bother writing them? And they are actually misleading.
3. if Eq.9, what is true rating r_j? Is it a non-personalized rating score?
4. In Eq. 10, If the two (R and w) are the same, why not considering merge them and only keeping one notation?

[1]Bias and Debias in Recommender System: A Survey and Future Directions .(https://arxiv.org/abs/2010.03240) Jiawei Chen, et.al.

Experimental design

The research questions are well-defined and rigorous investigation is performed. However, the method is not described in sufficient detail.
1. The baselines are not enough. The paper does not include any baslines from existing published works. I recommend the authors add three baselines for better comparison.
2. What are the Traditional statistical methods? please be more clear.
3. The proposed method without NMF, what do you mean? how do you predict the ratings? I think it is not trivial and you should not clearly point it .

Validity of the findings

All underlying data have been provided;
The novelty of the proposed method is trivial since using GCN and NMF is a commonly used method in recommendation. Please refer to the following for more work in this area.


[1]Bias and Debias in Recommender System: A Survey and Future Directions. (https://arxiv.org/abs/2010.03240)
[2]https://arxiv.org/abs/2305.14886
[3]https://arxiv.org/abs/2011.02260

---

## Round 0.2 · Minor Revisions

There is still room for further improvement. Some parts need further clarification. Presentation could be improved.

Reviewer 1 ·

Basic reporting

The questions and recommendations have been addressed and revised

Experimental design

The questions and recommendations have been addressed and revised

Validity of the findings

The questions and recommendations have been addressed and revised

Reviewer 2 ·

Basic reporting

This paper put forward a hybrid model with the existed GCN and NMF to deal with the user bias in the children's book rating data, and verified in two datasets that created by themselves.

Experimental design

The authors have conducted some experiments to verify the method. Maybe the datasets were created by themselves, so they have not compared with other methods that use GCN or NMF, too.

Validity of the findings

Since GCN and NMF are the existed methods, so the novelty maybe that they use the method to the children's book rating data.

Reviewer 3 ·

Basic reporting

I suggest the author reorganize Sections 2 and 3. Below are the reasons.

1. There are still too many repeated equations. e.g, ( Eq.1 and Eq.5,) ( Eq.2 ,Eq.10 and Eq.15), ( Eq.18 and Eq.13). ( Eq.16 Eq.17 and Eq.19). It is a weird thing and It can make the logic flow of this work very broken. Besides, it will also make it hard for readers to know what you proposed although Algorithm 1&2 helps. Please try to merge the repeated equations and make the logical flow more fluent.

2. It would be better to separate the preliminaries from the proposed method. I suggest the authors explicitly state what is preliminary, what is proposed by existing works, and what is proposed by them. For example, 1. Please use a separate section for the preliminaries. 2. What is the preliminaries for this work? Is Sec 2.3 also a preliminary? It seems that only Section 2.6 is the proposed method and all others are preliminaries. I still suggest that the author remove eq.6-9 if they are not used or put them in preliminaries. And only put what is truly proposed by the authors in the Proposed Method section.

Besides, the authors add a table i.e., Table 2 to specify the data statistics but did not give a detailed description of it. For example, what is the Number in the table?

Experimental design

no comment

Validity of the findings

no comment

Additional comments

My previous concerns about the draft are mostly addressed.

---

## Round 0.3 · Minor Revisions

Besides the issues raised by the reviewer, I suggest the authors conduct a double check of the whole paper.

**Language Note:** The Academic Editor has identified that the English language must be improved. PeerJ can provide language editing services - please contact us at copyediting@peerj.com for pricing (be sure to provide your manuscript number and title). Alternatively, you should make your own arrangements to improve the language quality and provide details in your response letter. – PeerJ Staff

Reviewer 2 ·

Basic reporting

no comment

Experimental design

no comment

Validity of the findings

no comment

Additional comments

The authors have answered the questions seriously, and revised the paper carefully, there has no more comment.

Reviewer 3 ·

Basic reporting

NA

Experimental design

NA

Validity of the findings

NA

Additional comments

Most of my concerns have been addressed. Please do a proofreading of the draft. For example, in Line 238, there seems an typo i.e., (??). BTW, I still did not get why Eq. 5~8 exist if they are not used in the proposed algorithm. There is no sign that they are used in Algorithm 1 or 2. However, Eq. 5~8 are used to mitigate the bias. If they are not used in the proposed algorithm, which part of the proposed algorithm are dealing with the bias? It is still hard for me understand the overall logic here.

---

## Round 0.4 · accepted · Accept

All reviewers' comments have been well addressed. It is now ready for publication.